# Low dose cisplatin weekly versus high dose cisplatin every three weeks in primary chemoradiotherapy in head and neck cancer patients with low skeletal muscle mass: The CISLOW-study protocol

**Anouk W. M. A. Schaeffers**[1], **Lot A. Devriese**[2], **Carla H. van Gils**[3], **Jan Willem Dankbaar**[4], **Jens Voortman**[5], **Jan Paul de Boer**[6], **Marije Slingerland**[7], **Mathijs P. Hendriks**[8], **Ernst J. Smid**[9], **Geert W. J. Frederix**[3], **Remco de Bree**[1] *

1 Department of Head and Neck Surgical Oncology, University Medical Center Utrecht, Utrecht, The Netherlands, 2 Department of Medical Oncology, University Medical Center Utrecht, Utrecht, The Netherlands, 3 Julius Center for Health Sciences and Primary Care, University Medical Center Utrecht, Utrecht, The Netherlands, 4 Department of Radiology, University Medical Center Utrecht, Utrecht, The Netherlands, 5 Department of Medical Oncology, Amsterdam UMC, Cancer Center Amsterdam, location VUmc, Amsterdam, The Netherlands, 6 Department of Medical Oncology, Antoni van Leeuwenhoek, Amsterdam, The Netherlands, 7 Department of Medical Oncology, Leiden University Medical Center, Leiden, The Netherlands, 8 Department of Medical Oncology, Northwest Clinics, Alkmaar, The Netherlands, 9 Department of Radiotherapy, University Medical Center Utrecht, Utrecht, The Netherlands

* r.debree@umcutrecht.nl

## Abstract

Chemoradiotherapy with cisplatin in a triweekly regimen of 100 mg/m2 body surface area, is used to treat locally advanced head and neck squamous cell carcinoma (HNSCC) with curative intent. Cisplatin dose limiting toxicity (CDLT) occurs often and impedes obtaining the planned cumulative cisplatin dose. A cumulative cisplatin dose of 200 mg/m2 or more is warranted for better survival and locoregional control. Patients with a low skeletal muscle mass (SMM) have a three-fold higher risk of developing CDLT than patients with a normal SMM. SMM can be assessed through measurements on routinely performed diagnostic head and neck CT- or MRI-scans. A weekly regimen of 40 mg/m2 body surface area cisplatin is proposed as a less toxic schedule, which possibly decreases the risk of developing CDLT and enables reaching a higher cumulative cisplatin dose. The aim of this multicenter randomized clinical trial (NL76533.041.21, registered in the Netherlands Trial Register) is to identify whether a regimen of weekly cisplatin increases compliance to the planned chemotherapy scheme in HNSCC patients with low SMM. The primary outcome is the difference in compliance rate, defined as absence of CDLT, between low SMM patients receiving either the weekly or triweekly regimen. Secondary outcomes consist of toxicities, the cumulative cisplatin dose, time to recurrence, incidence of recurrence at two years of follow-up, location of recurrence, 2-year overall, disease free and disease specific survival, quality of life, patient's experiences, and cost-effectiveness.

**Funding:** The Netherlands Organization for Health Research and Development (ZonMw), project number 10140021910002. The funders did not and will not have a role in study design, data collection and analysis, decision to publish, or preparation of the manuscript.

**Competing interests:** The authors have declared that no competing interests exist.

## Introduction

Worldwide 900.000 new cases of head and neck squamous cell carcinoma (HNSCC) were estimated in 2018, which is approximately 5% of all new cancer cases [1]. Two-thirds of HNSCC patients present with advanced stage HNSCC. Cisplatin-based chemoradiotherapy (CCRT) is the preferred treatment for patients with unresectable or functionally irresectable HNSCC aiming for organ and function preservation. Next to CCRT, also other treatments that are possibly less invasive are currently available [2]. However, surgery will not be discussed in this article.

Cisplatin, dosed as 100 mg/m$^2$ body surface area (BSA), every three weeks for three cycles in combination with conventionally fractionated radiotherapy, consisting of 70 Gy in 35 fractions (five times a week) is the standard of care [3–9]. An alternative frequently used scheme is weekly cisplatin, dosed as 40 mg/m$^2$ BSA, for seven cycles with concurrent radiotherapy as described above. Cisplatin acts as a radiosensitizer. Locoregional control and survival improve significantly after treatment with CCRT compared to a single modality treatment with radiation [10–12]. CCRT is reserved only for patients who are deemed cisplatin fit, but nevertheless approximately one third of the patients is not able to complete the proposed triweekly regimen due to cisplatin dose limiting toxicity (CDLT). CDLT is caused by the detrimental side-effects of CCRT which are often comprised of nephrotoxicity, neurotoxicity, ototoxicity, leukopenia, mucositis and dermatitis [13–16].

HNSCC patients with low skeletal muscle mass (SMM) are more prone to developing CDLT [17–19]. In HNSCC patients SMM can be determined by delineating the cross-sectional muscle area (CSMA) by semi-automatic segmentation of the skeletal muscles at the third cervical vertebra (C3), using computed tomography (CT) or magnetic resonance imaging (MRI) scans. Subsequently CSMA at the (most frequently used) third lumbar vertebra (L3) can be calculated using a formula with age, weight, length, sex and the CSMA as variables. Then the lumbar skeletal muscle index (LSMI), the most often used measure for SMM, can be calculated by correcting for length [20–24]. The association between low SMM and CDLT might be explained by hydrophilic characteristics of cisplatin, causing distribution of cisplatin to fat-free body mass which is mainly composed of SMM [25]. Since cisplatin dosage is calculated using BSA, especially patients with low SMM but a large BSA receive relatively large quantities of cisplatin, possibly leading to a higher peak dosage. A higher peak dosage of cisplatin might consequently cause a higher rate of toxicities [26].

A weekly regimen of 40 mg/m$^2$ cisplatin for seven weeks might be an excellent alternative to the triweekly regimen in order to decrease rates of early treatment cessation, but studies show conflicting results [13, 15, 27–33]. Noronha et al. reported superiority of triweekly 100 mg/m$^2$ cisplatin compared to weekly cisplatin dosed as 30 mg/m$^2$ in cumulative two years loco-regional control [28]. Szturz et al. found no statistical difference in survival between patients treated with triweekly 100 mg/m$^2$ cisplatin versus weekly 40 mg/m$^2$ cisplatin [13]. The European Society for Medical Oncology recommends the triweekly 100 mg/m$^2$ cisplatin regimen (level 1 evidence), over the weekly 40 mg/m$^2$ cisplatin regimen (level 2 evidence) [8]. Early discontinuation of therapy automatically leads to a reduced cumulative cisplatin dose, which is associated with lower overall survival. Therefor it can be anticipated that particularly patients with low SMM might benefit from weekly low dose cisplatin based concurrent CRT to achieve an adequate cumulative dose comparable to patients with normal SMM [10, 38, 40, 41].

## Materials and methods

The multicenter randomized clinical trial CISLOW is approved by the Medical Ethics Committee (METC), registered in the Netherlands Trial Register (NL76533.041.21), funded by The

Netherlands Organisation for Health Research and Development (ZonMw; 10140021910002) and conducted by five head and neck centers of the Dutch Head and Neck Society: University Medical Center Utrecht (UMCU), Amsterdam University Medical Centers location VUmc, Antoni van Leeuwenhoek, Leiden University Medical Center and Northwest Clinics. The SPIRIT schedule is shown in Fig 1. The starting date was January 2022. HNSCC patients who

| | | | | STUDY PERIOD | | | | | | |
|---|---|---|---|---|---|---|---|---|---|---|
| | Enrolment | Before randomization | Allocation | Intervention<br>Cisplatin-based CRTa | Follow-up (in months after end of CRTa) | | | | | |
| TIMEPOINT** | -t₁ | | 0 | 0 – 7 weeks | 3 | 6 | 12 | 18 | 24 | |
| **ENROLMENT:** | | | | | | | | | | |
| Eligibility screen | X | | | | | | | | | |
| Informed consent | X | | | | | | | | | |
| Allocation | | | X | | | | | | | |
| **INTERVENTIONS:** | | | | | | | | | | |
| *Low SMMᵇ weekly cisplatin* | | | | ⟷ | | | | | | |
| *Low SMMᵇ triweekly cisplatin* | | | | ⟷ | | | | | | |
| *Normal SMMᵇ local standard of care cisplatin* | | | | ⟷ | | | | | | |
| **ASSESSMENTS:** | | | | | | | | | | |
| *Demographic data* | X | | | | | | | | | |
| *SMMᵇ measurement* | | X | | | | X | | X | | |
| *Adverse events* | | | | X | X | X | X | X | X | |
| *Dose limiting toxicity* | | | | X | | | | | | |
| *Quality of life questionnaires* | | | X | | X | X | X | | X | |
| *Semi-structured interviews in low SMM patients in UMCUᶜ* | | | | | X | | | | | |
| *Oncologic outcomes (survival, recurrence)* | | | | | X | X | X | X | X | |

**Fig 1. SPIRIT schedule of enrollment, interventions and assessments.** a) CRT = Chemoradiotherapy, b) SMM = skeletal muscle mass, c) UMCU = University Medical Center Utrecht.

**Table 1. Inclusion and exclusion criteria.**

| Inclusion criteria | Exclusion criteria |
|---|---|
| The patient is considered, eligible and planned for primary cisplatin-based chemoradiotherapy by treating physician; | The patient is mentally disabled and is unable to give informed consent; |
| The patient is eighteen years of age or older; | The patient has an absolute contraindication for triweekly cisplatin (100 mg/m$^2$) or is not cisplatin fit according to the treating physician; |
| The patient has sufficient understanding of Dutch and medical consequences to give informed consent. | The patient receives cisplatin in a non-primary intent or as induction treatment; |
| | The interval between the diagnostic scan used for skeletal muscle mass assessment and start of chemoradiotherapy is more than two months; |
| | An accurate skeletal muscle mass measurement was not possible*. |

* Bilateral lymph node metastasis or artefacts at the level of the third cervical vertebra can compromise accurate skeletal muscle mass assessment. If imaging of the third lumbar vertebra is available and without artefacts, skeletal muscle mass can be assessed on this slice.

are planned for CCRT, proposed by a multidisciplinary tumor board, and meet the criteria (as described in Table 1) will be asked to participate in this study.

## Assessment of skeletal muscle mass

SMM will be assessed on a routinely performed CT- or MRI-scan of the head and neck as shown in Figs 2 and 3. The SMM will be estimated based on measurement of the cross-sectional area of the sternocleidomastoid muscle and paravertebral muscles on the level of C3. Choosing a specific image slice at the level of C3 will be done by selecting the first slice showing both transverse processes and the entire vertebral arc when scrolling from caudal to cranial direction. Delineation of muscle tissue will be performed using the Slice-O-matic software v5.0 (Tomovision, Canada) [20]. If lymph node metastasis prohibits accurate delineation of either sternocleidomastoid muscle, the CSMA of the contralateral sternocleidomastoid muscle will be segmented and used to estimate the CSMA of the affected sternocleidomastoid muscle. In CT-scans muscle tissue will be identified semi-automatically using Hounsfield Unit range settings from -29 to +150, which is specific for muscle tissue [34]. In MRI-scans the muscle tissue will be manually segmented. The overall intraclass correlation coefficient for the CSMA obtained by CT and MRI has shown to be excellent (intraclass correlation coefficient 0.9, p<0.01), and can therefore be used interchangeably for measuring CSMA at the level of C3 [35]. Subsequently, the cervical skeletal muscle index (CSMI) will be calculated by dividing the CSMA at the level of C3 by the squared height of the patient. The LSMI will be calculated as following [20]:

$$\begin{aligned}&\text{CSMA at third lumbar vertebra (cm2)}\\&= 27.304 + 1.363 * \text{CSMA at C3 (cm2)} - 0.671 * \text{Age (years)} + 0.640 * \text{weight (kg)}\\&\quad + 26.442 * \text{Sex (female}\\&= 1,\ \text{male} = 2)\end{aligned}$$

By dividing the CSMA at the level of the third lumbar vertebra (L3) by the squared height of the patient, the LSMI is calculated.

This method has been validated and appeared to be robust when comparing it to the common method measuring the muscle area at the level of the third lumbar vertebrae (L3) [22, 23, 34, 35].

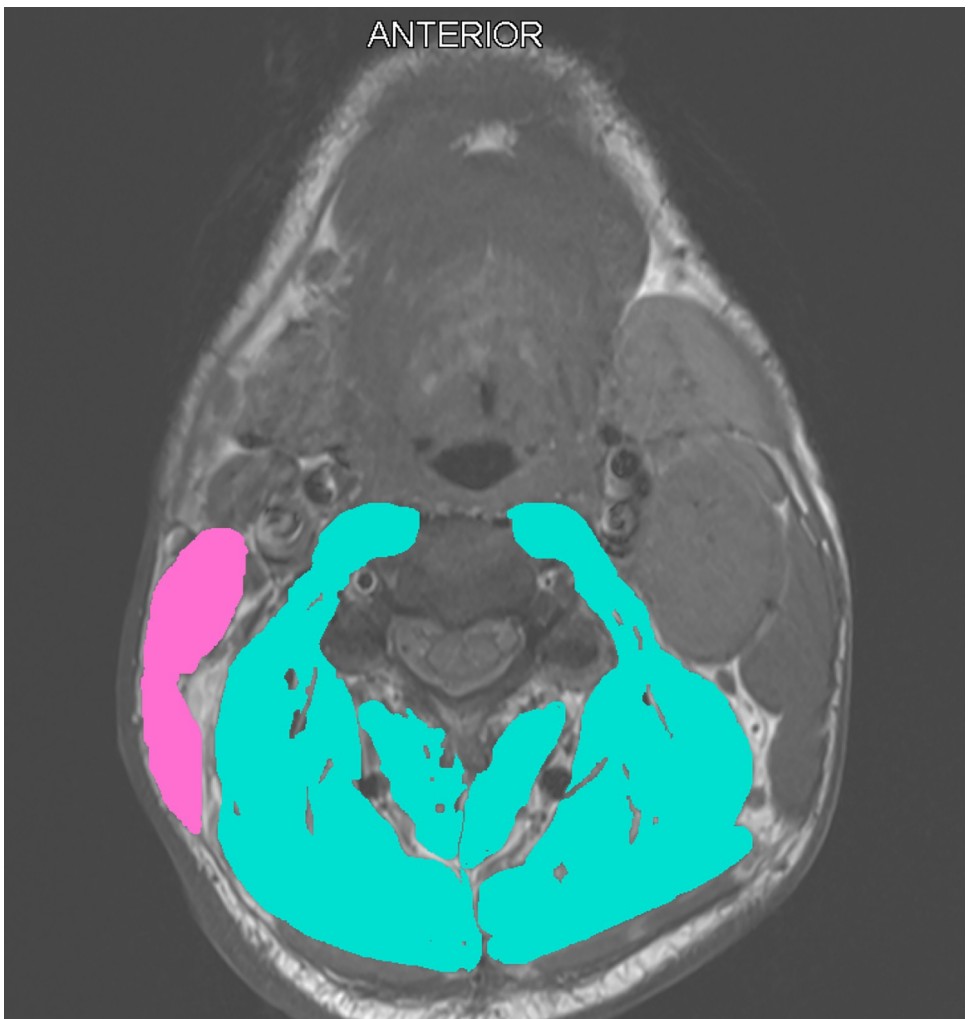

**Fig 2. Cross sectional muscle area of cervical skeletal muscles in a patient without low skeletal muscle mass.** The right sternocleidomastoid muscle area is used twice to account for lymph node metastasis on the left side.

## Intervention

Patients with low SMM (LSMI $\leq$ 43.2 cm$^2$/m$^2$) will be randomized between weekly cisplatin (40 mg/m$^2$) and triweekly cisplatin (100 mg/m$^2$) and patiens with a normal SMM (LSMI > 43.2 cm$^2$/m$^2$) will receive the local standard of care, which is triweekly cisplatin in the UMCU, Amsterdam University Medical Centers location VUmc, Leiden University Medical Center and Northwest Clinics and weekly cisplatin in Antoni van Leeuwenhoek, and will be observed [17, 36].

## Outcomes

The primary outcome of this study is the difference in compliance (defined as absence of CDLT) rate to the proposed cisplatin scheme between weekly cisplatin (40 mg/m$^2$) and triweekly cisplatin (100 mg/m$^2$) in patients with low SMM. CDLT for the triweekly schedule is defined as any toxicity resulting in a cisplatin dose-reduction of $\geq$50%, a postponement of treatment of $\geq$4 days or a definite termination of cisplatin after the first or second cycle of therapy, which is in line with the study of Wendrich et al. [17]. For weekly cisplatin regimens

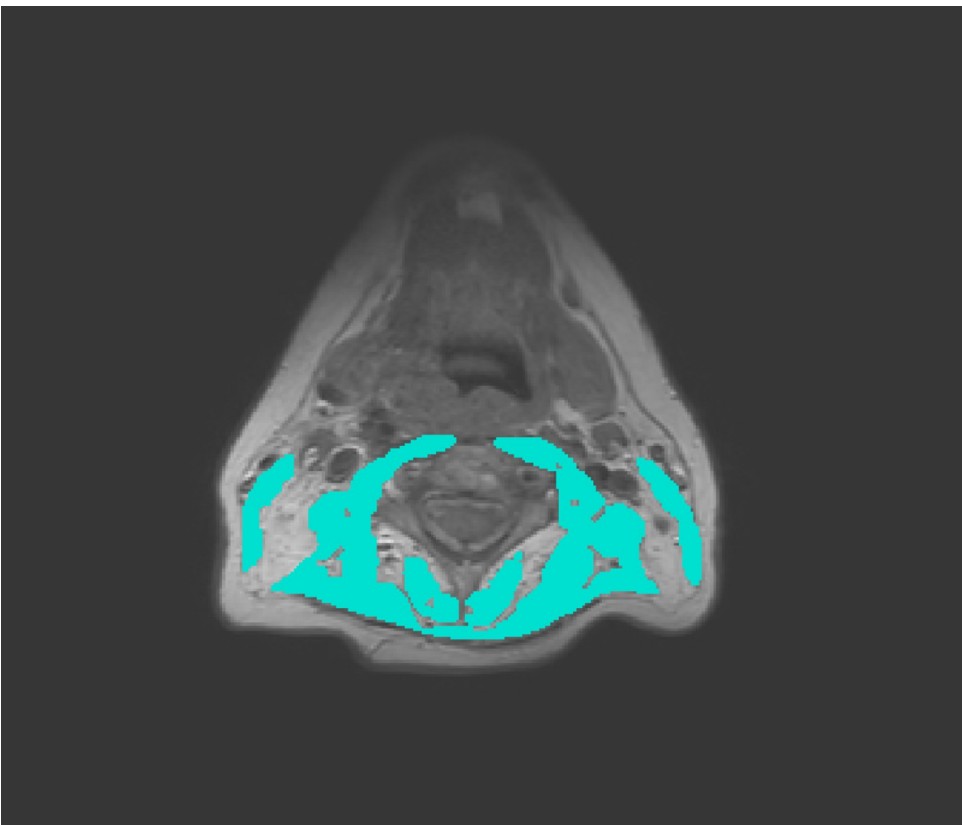

**Fig 3. Cross sectional muscle area of cervical skeletal muscles in a patient with low skeletal muscle mass.**

CDLT is defined as skipped cisplatin treatments or a definite termination of cisplatin before all scheduled cycles of therapy are given. Secondary outcomes of this study are treatment cumulative cisplatin dose, toxicities according the Common Toxicity Criteria Adverse Events version 5 during treatment and for a period of 24 months from the end of treatment, time to recurrence, incidence of recurrence at two years follow-up, location of recurrence (e.g., local, regional and distant recurrence), 2-years survival, quality of life measured by questionnaires (EORTC QLQ-C30, EORTCQLQ-H&N35, EQ-5D-5L) before and 3, 6, 12 and 24 months after CCRT, and a qualitative analysis using semi-structured interviews to assess patients' experiences towards the treatment schedule [14]. Cost-effectiveness analysis will be performed to compare triweekly and weekly cisplatin treatment in low SMM patients, with difference in costs, impact on quality of life and incremental cost-effectiveness ratio as outcomes. Electronic patient records will be checked during follow-up for two years. Follow-up will be according to Dutch guidelines [37]. Data management will be done using Castor software, taking privacy regulations into account. Patients' are insured by a trial subject insurance, the protocol has been checked by the Medical Research Ethics Committee in Utrecht (number 21–329), serious adverse events will be reported to the Medical Research Ethics Committee conform protocol and an external monitor checks the study team repeatedly for correct protocol handling.

## Sample size calculation and statistical procedures

Sample size calculation is based on the following assumptions. For the control arm of patients with low SMM and a triweekly regimen we expect a compliance rate of 55.7%, as was observed

in the study of Wendrich et al. in a similar population [17]. In the intervention arm of patients with low SMM and weekly cisplatin, we expect a compliance rate of 86.3%. This proportion was observed in the study of Wendrich et al. for patients with normal SMM and triweekly cisplatin [17]. Our hypothesis is that this same compliance rate can be obtained by using weekly cisplatin in patients with low SMM. The sample size was calculated to be able to indicate the difference between 55.7% and 86.3% as statistically significant, using a two-sided alpha of 0.05 and a power of 85%. With an expected drop-out rate of 5% and a low SMM incidence rate of approximately 55%, 129 patients must be included [17] Patients with low SMM are centrally randomized 1:1 for triweekly high dose or weekly low dose cisplatin. Allocation is performed by a central office on-site computer combined with allocations kept in a locked, unreadable computer file that investigators can assess only after the characteristics of an enrolled patients are entered. A stratified permuted-block procedure randomizes patients to the groups on a 1:1 ratio. Strata comprise center and tumor stage (UICC I-III and IV) [38]. Neither patients, investigators nor office personnel are blinded to the group chosen by the randomization procedure. Fisher's exact tests, Pearson Chi square tests, independent sample t-tests and Mann-Whitney U tests will be used to assess group differences. All analyses will be two-sided and $p < 0.05$ is considered significant. Missing data will be handled using multiple imputation, if needed; we only expect few missing data to occur due to the nature and scale of the study.

## Results and discussion

Cisplatin is routinely used in the curative treatment of locally advanced HNSCC as a radiosensitizing agent. Among various proposed treatment schedules, differing in frequency, dose, and route of administration, there is level 1 evidence for significant improvement in locoregional control and overall survival achieved by triweekly high dose (three cycles of 100mg/m$^2$) intravenous cisplatin given concurrently with conventional external beam radiotherapy compared to radiotherapy alone [8]. Despite indisputable efficacy, high rates of severe acute and late adverse events remain of concern. In fact, due to unacceptable systemic and local toxicities, up to 40% of patients fail to comply to all three planned cycles of high-dose cisplatin, which decrease local control and overall survival (3-year overall survival from 72% to 52% in HPV-negative patients) in individual patients [13, 39]. Besides, severe late toxicity (which has been reported in up to 13% of patients) is of concern leading to permanent morbidity [13, 40].

A systematic review pointed out a significant association between total cisplatin dose and overall survival [41]. Also Spreafico et al. found in a pooled analysis of 659 patients with locally advanced HNSCC with HPV positive tumors a survival benefit with cisplatin above 200mg/m$^2$ [42]. In a systematic review Szturz et al. compared the standard, triweekly high dose (three cycles of 100mg/m$^2$) cisplatin, and its alternative, weekly low dose (seven cycles of 40mg/m$^2$) cisplatin with concurrent radiation in locally advanced HNSCC. They found that, although treatment adherence was higher in the weekly schedule (88%) compared to the triweekly schedule (71%; p = 0.0017), no statistical difference in survival was observed for the total population [13]. Although Jacinto et al. also concluded in a systematic review that triweekly cisplatin schedule should remain the standard of care for locally advanced HNSCC, they suggest a specific subset of not yet identified patients in whom weekly concurrent cisplatin is more appropriate to achieve a higher cumulative cisplatin dose [43].

Across cancer types, patients with a low SMM have a significantly higher risk of severe toxicity (Odds ratio (OR) 2.19–4.08) and dose-limiting toxicity (OR 2.24–2.48) compared to patients without low SMM [44, 45]. Patients with HNSCC and a low SMM show a higher risk CDLT. In a retrospective study, Wendrich et al. were the first to show an association between low SMM and the occurrence of CDLT in patients with HNSCC treated with triweekly

cisplatin-based CCRT. CDLT was defined as any toxicity resulting in a chemotherapy dose-reduction of $\geq$50%, a postponement of treatment of $\geq$4 days or a definite termination of chemotherapy after the first or second cycle of therapy. Of 112 patients included, 30.4% experienced CDLT. The SMM was estimated based on the same validated method we use in the CISLOW study [20, 23]. The optimal cut-off value for low SMM as a predictor of CDLT was $\leq$43.2cm$^2$/m$^2$. Using this cut-off, 54.5% patients had low SMM. Patients with low SMM experienced CDLT more frequently than patients with normal SMM (44.3% vs. 13.7%, p<0.001) and received a higher dose of chemotherapy per kg lean body mass (estimated from SMM, p = 0.044). Low SMM was independently inversely associated with CDLT (OR 0.93, 95%CI: 0.88–0.98) and patients experiencing CDLT showed shorter overall survival rates than patients who did not experience CDLT (mean 36.6 vs. 54.2 months, p = 0.038) [17]. Recently, this study was enlarged (n = 343), including patients whom were treated in a period of 10 years.

In line with the results from Wendrich et al., patients with low SMM prior CCRT were more likely to develop CDLT compared to patients without low SMM at diagnosis (51.8% vs. 36.8%, p<0.01) [17, 46]. These findings were confirmed by Bril et al. in a cohort of 153 HNSCC patients treated by chemoradiotherapy with triweekly high dose cisplatin in AvL. Of the 153 included patients 84 had low SMM (54.9%) defined as low CSMA at level C3 based on a cut-off value of 10.7 cm$^2$ for women and 13.1 cm$^2$ for men [18]. CDLT was defined as any toxicity leading to a cumulative cisplatin dose less than 200 mg/m$^2$. In total 37 (24.2%) patients experienced CDLT, and patients with low SMM experienced significantly more CDLT compared to patients without low SMM (33.7% vs. 10.9%, p = 0.001), which is in line with other studies [18, 19, 47, 48]. Low SMM (HR 3.6, 95% CI 1.5–8.7, p = 0.004) and an estimated glomerular filtration rate of 60–70 mL/min (HR 4.2, 95% CI 1.4–12.7, p = 0.012) were independent predictors for CDLT in multivariate analysis. Moreover a significant lower OS was observed in patients who experienced CDLT compared to patients who did not experience CDLT (26.3 vs. 40.7 months, p = 0.003) [18]. Although these studies showed that HNSCC patients with low SMM have a higher risk of CDLT, the results are difficult to compare because the definitions of CDLT and low SMM vary [18, 46].

In the CISLOW study, we use the same CDLT definition and cut-off point for low SMM as Wendrich et al. did since they performed their study in the same type of population [17]. Moreover, this definition takes the importance of the cumulative cisplatin dose into account. There remains some controversy about the optimal cut-off to define a low SMM and most are not validated in the HNSCC population. Often used cut-off values used are defined by Prado et al. and Martin et al. but these studies are executed in patients with lung or gastrointestinal malignancies and SMM is assessed at the level of L3 instead of C3 [25, 49]. Also in HNSCC studies different cut-off values are found when addressing different calculating methods or outcome values: for example Zwart et al. defined different cut-off points for different kinds of adverse events and Chargi et al. proposed to define low SMM based on the mean minus two standard-deviations, after grouping based on gender and BMI [50, 51]. Since Wendrich et al. specifically created a cut-off point for patients with HNSCC receiving chemoradiotherapy in the Netherlands, based on CDLT, using their cut-off point in our study with the same population is plausible [17]. In the CISLOW study we have chosen to stratify for the probably most important factors center and tumor stage. As HPV status is enclosed in the tumor stage, this will imply stratification for HPV status as well.

SMM assessment will be performed on CT- or MRI-scans, depending on availability. The whole body total skeletal muscle volume is often approached by assessing the CSMA at the level of L3 using a CT-scan and adjusting this for patient's height leading to the skeletal muscle index [24, 25]. In the diagnostic work up for HNSCC an abdominal scan is not standard of care, but Swartz et al. created a method to assess SMM on the level of C3, with a good

correlation (r = 0.785) [20]. This was validated by Bril et al. and multiple studies showed excellent intra-observer and inter-observer agreement, even when using slightly different formulas [20, 23, 52–55]. Sometimes no CT-scan but an MRI-scan is made in the diagnostic work-up. Since there is a strong correlation between the CSMA measured at the level of C3 using a CT- and MRI-scan (intraclass correlation coefficient 0.9, $p<0.01$) and an excellent intra-observer agreement was found, we find it justifiable to use CT-scans or MRI-scans interchangeably in assessing SMM [22, 35, 53, 56].

In short, CDLT is common during CCRT in patients with HSNCC and consequently leads to a lower cumulative dosage of cisplatin which subsequently can affect survival [10, 17–19]. A solution to decrease the occurrence of CDLT while maintaining adequate cisplatin levels is warranted. By selecting a subgroup of patients that are especially prone to CDLT, namely patients with low SMM, and randomizing these patients between the triweekly and a the weekly regimen of cisplatin, we believe we can investigate whether the weekly regimen might be this solution. The results of this trial can lead to a change in guidelines for cisplatin treatment, might improve our understanding of the effect of SMM on CDLT and enhance personalized treatment in patients with HNSCC.

## Conclusions

Low SMM is predictive for CDLT in HNSCC patients receiving triweekly cisplatin in combination with radiotherapy with curative intent. Weekly cisplatin dosed as 40 mg/m$^2$ BSA seems less toxic and is potentially non-inferior in comparison to triweekly cisplatin dosed as 100 mg/m$^2$. The aim of the CISLOW-study is to investigate whether the compliance of HNSCC patients with low SMM to cisplatin treatment as part of concurrent chemoradiation can be improved by using a weekly cisplatin regimen as high as HNSCC patients without a low SMM receiving a triweekly cisplatin regimen.

## Supporting information

**S1 Checklist. SPIRIT checklist.**
(DOCX)

**S2 Checklist. STROBE statement—checklist of items that should be included in reports of observational studies.**
(DOCX)

**S1 File. Informed consent materials in Dutch.**
(PDF)

**S2 File. Monitoring plan.**
(PDF)

**S3 File. Data management plan.**
(PDF)

**S4 File.**
(PDF)

## Author Contributions

**Conceptualization:** Anouk W. M. A. Schaeffers, Lot A. Devriese, Carla H. van Gils, Geert W. J. Frederix, Remco de Bree.

**Funding acquisition:** Remco de Bree.

**Investigation:** Anouk W. M. A. Schaeffers.

**Methodology:** Anouk W. M. A. Schaeffers, Lot A. Devriese, Carla H. van Gils, Geert W. J. Frederix, Remco de Bree.

**Project administration:** Anouk W. M. A. Schaeffers.

**Resources:** Anouk W. M. A. Schaeffers, Lot A. Devriese, Jan Willem Dankbaar, Jens Voortman, Jan Paul de Boer, Marije Slingerland, Mathijs P. Hendriks, Ernst J. Smid, Remco de Bree.

**Supervision:** Lot A. Devriese, Carla H. van Gils, Geert W. J. Frederix, Remco de Bree.

**Writing – original draft:** Anouk W. M. A. Schaeffers.

**Writing – review & editing:** Lot A. Devriese, Carla H. van Gils, Jan Willem Dankbaar, Jens Voortman, Jan Paul de Boer, Marije Slingerland, Mathijs P. Hendriks, Ernst J. Smid, Geert W. J. Frederix, Remco de Bree.

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
