## [Decision Letter · Decision Letter 0]

28 Sep 2023

PONE-D-23-16285Low dose cisplatin weekly versus high dose cisplatin every three weeks in primary  in head and neck cancer patients with low skeletal muscle mass: the CISLOW-study protocolPLOS ONE

Dear Dr. de Bree,

Thank you for submitting your manuscript to PLOS ONE. After careful consideration, we feel that it has merit but does not fully meet PLOS ONE’s publication criteria as it currently stands. Therefore, we invite you to submit a revised version of the manuscript that addresses the points raised during the review process.

We look forward to receiving your revised manuscript.

Kind regards,

Antonino Maniaci

Academic Editor

PLOS ONE

Journal Requirements:

2. In the Methods section of your revised manuscript, please include the tentative/proposed date when the study will be initiated.

"The funders did not and will not have a role in study design, data collection and analysis, decision to publish, or preparation of the manuscript."

5. We note that the original protocol that you have uploaded as a Supporting Information file contains an institutional logo. As this logo is likely copyrighted, we ask that you please remove it from this file and upload an updated version upon resubmission.

Additional Editor Comments:

One reviewer suggested maior improvements, please perform as required. Bests

Reviewers' comments:

Reviewer's Responses to Questions

**Comments to the Author**

1. Does the manuscript provide a valid rationale for the proposed study, with clearly identified and justified research questions?

Reviewer #1: Yes

Reviewer #2: Yes

2. Is the protocol technically sound and planned in a manner that will lead to a meaningful outcome and allow testing the stated hypotheses?

Reviewer #1: Yes

Reviewer #2: Yes

3. Is the methodology feasible and described in sufficient detail to allow the work to be replicable?

Reviewer #1: Yes

Reviewer #2: Yes

4. Have the authors described where all data underlying the findings will be made available when the study is complete?

Reviewer #1: Yes

Reviewer #2: Yes

5. Is the manuscript presented in an intelligible fashion and written in standard English?

Reviewer #1: Yes

Reviewer #2: Yes

6. Review Comments to the Author

You may also provide optional suggestions and comments to authors that they might find helpful in planning their study.

Reviewer #1: maior concerns before acceptation:

Introduction:

- The introduction provides good background and rationale for the study, clearly highlighting the problem of chemotherapy toxicity in HNSCC patients.

- The writing is generally clear. Consider reducing the length of some sentences for easier readability.

- The study objectives are clearly stated at the end.

- line 11, alternative less invasive surgical methods as robotic surgery for head and neck or oropharyngeal cancer should be considered. pleae discuss and cite doi:10.1002/rcs.2106

- line 2, a study reported in patients with local control, values of pre-treatment medium and minimum ADC were lower than ADC values of patients with persistent or recurrent disease, with values, respectively, of 0.83 ± 0.02 × 10-3 mm2/s and 0.59 ± 0.02 × 10-3 mm2/s (vs 0.94 ± 0.05 × 10-3 mm2/s and 0.70 ± 0.05 × 10-3 mm2/s). ROC curve analysis displayed statistical significance as regarding the medium ADC value, showing a sensitivity of 50% and a specificity of 84.8%. ROC analysis of the values minimum ADC showed a sensitivity of 42.9% and specificity of 87.9%. , discuss and cite doi: 10.1007/s11547-017-0733-y.

- also reconstructive surgery for t2-t3 tumours should be discussed because avoid the use of tracheostomy and allow organ function. please discuss and cite doi:10.23812/19-282-L.

Methods:

- The methods section covers all key elements - study design, participants, interventions, outcomes, sample size, randomization, etc.

- Subheadings would help organize the different components of the methods.

- Avoid excessive use of abbreviations on first mention - write out full terms.

- Provide a bit more details on the statistical analysis plan for each aim.

Results:

- As this is a protocol paper, there are no results yet. Consider adding a section describing the expected results/analysis.

Discussion:

- The discussion provides good context and rationale for the study design choices.

- References are appropriate. Limitations of the study design are acknowledged.

-Splitting some lengthy paragraphs into two would further enhance readability.

Reviewer #2: I read with great interest the CISLOW study protocol by Schaeffers et al. on low dose cisplatin weekly versus high dose cisplatin every three weeks in primary chemoradiotherapy in head and neck cancer patients with low skeletal muscle mass. The protocol is sound and original, the rationale of the research question is valid and the methodology feasible and described in sufficient detail to allow the work to be replicable. Therefore, I would suggest to accept it without further revisions.

7. PLOS authors have the option to publish the peer review history of their article (what does this mean?). If published, this will include your full peer review and any attached files.

Reviewer #1: No

Reviewer #2: No

---

## [Author Response · Author response to Decision Letter 0]

5 Oct 2023

 We have changed the layout by adding headers level one in the manuscript.

2. In the Methods section of your revised manuscript, please include the tentative/proposed date when the study will be initiated.

 We have added this in line 78: ‘The starting date was January 2022’.

 As far as we are concerned we mentioned ZonMw at all places required, as our funder. We would like to know what not matches and whether we should change anything else. We could not find this out on the application form. If you want the statements of us in any other way than in this rebuttal letter, we would like to be informed where specifically have to make changes. 

4. Thank you for stating the following financial disclosure: "The funders did not and will not have a role in study design, data collection and analysis, decision to publish, or preparation of the manuscript." At this time, please address the following queries:

 ZonMw, a Dutch governmental funding institution, granted us financial support. 

 Hereby we state that the funders had no role in study design, data collection and analysis, decision to publish, or preparation of the manuscript.

Mrs. Schaeffers and mr. Frederix salaries are both (partly) paid by the ZonMw grant.

d) If you did not receive any funding for this study, please state: “The authors received no specific funding for this work.” Please include your amended statements within your cover letter; we will change the online submission form on your behalf.

 We did receive funding for this study so we cannot state this. 

5. We note that the original protocol that you have uploaded as a Supporting Information file contains an institutional logo. As this logo is likely copyrighted, we ask that you please remove it from this file and upload an updated version upon resubmission.

As far as we can see, our protocol does not contain any institutional logo. The informed consent material and monitoring plan however do, so we have changed these documents and uploaded these without logo into your submission system. 

1. Reviewer 1 Comment 1: The introduction provides good background and rationale for the study, clearly highlighting the problem of chemotherapy toxicity in HNSCC patients.

Thank you very much for your kind comment.

2. Reviewer 1 Comment 2: The writing is generally clear. Consider reducing the length of some sentences for easier readability.

We agree with you, we have shortened several sentences in the introduction by splitting up sentences in line 36-37, 40-41 and 66-67.

3. Reviewer 1 Comment 3: The study objectives are clearly stated at the end.

Thank you, we did not change this part.

4. Reviewer 1 Comment 4: line 11, alternative less invasive surgical methods as robotic surgery for head and neck or oropharyngeal cancer should be considered. pleae discuss and cite doi:10.1002/rcs.2106

We have added the following to our manuscript, in line 32: ‘Next to CCRT, also other treatments that are possibly less invasive are currently available.[your reference]. However surgery will not be discussed in this article.’

5. Reviewer 1 Comment 5: line 2, a study reported in patients with local control, values of pre-treatment medium and minimum ADC were lower than ADC values of patients with persistent or recurrent disease, with values, respectively, of 0.83 ± 0.02 × 10-3 mm2/s and 0.59 ± 0.02 × 10-3 mm2/s (vs 0.94 ± 0.05 × 10-3 mm2/s and 0.70 ± 0.05 × 10-3 mm2/s). ROC curve analysis displayed statistical significance as regarding the medium ADC value, showing a sensitivity of 50% and a specificity of 84.8%. ROC analysis of the values minimum ADC showed a sensitivity of 42.9% and specificity of 87.9%. , discuss and cite doi: 10.1007/s11547-017-0733-y.

Since our protocol is not about the predictive value of specific tumor characteristics, such as the apparent diffusion coefficient of tumors, that can be visualized using imaging, we believe that the additional information you want us to add to our protocol, does not improve the comprehensibility of the protocol. 

6. Reviewer 1 Comment 6: also reconstructive surgery for t2-t3 tumours should be discussed because avoid the use of tracheostomy and allow organ function. please discuss and cite doi:10.23812/19-282-L.

As for comment 5, we believe that the clarity of our manuscript remains more intact if we do not add additional information regarding surgical reconstructive options, since the scope of our research is about primary chemoradiotherapy. Therefore we did not add the above suggested citation to our manuscript. 

7. Reviewer 1 Comment 7: The methods section covers all key elements - study design, participants, interventions, outcomes, sample size, randomization, etc.

Thank you, we did not change this part.

8. Reviewer 1 Comment 8: Subheadings would help organize the different components of the methods.

We have added several subheadings to the method section to increase readability, namely ‘assessment of skeletal muscle mass’, ‘intervention’, ‘ouctomes’ and ‘sample size calculation and statistical procedures’. 

9. Reviewer 1 Comment 9: Avoid excessive use of abbreviations on first mention - write out full terms.

We removed the ESMO abbreviation in line 64.

We removed the abbreviation NWHHT in line 76. We removed the abbreviations of the different sites (except our own site since the abbreviation is mentioned more often) in line 76-78 and line 114. We removed the abbreviation HU in line 90. We removed the abbreviation ICC in line 100. We removed the abbreviation CTCAE in line 128.

10. Reviewer 1 Comment 10: Provide a bit more details on the statistical analysis plan for each aim.

We added the following text to line 16: ‘Fisher’s exact tests, Pearson Chi square tests, independent sample t-tests and Mann-Whitney U tests will be used to assess group differences. All analyses will be two-sided and p<0.05 is considered significant. Missing data will be handled using multiple imputation, if needed; we only expect few missing data to occur due to the nature and scale of the study.’

11. Reviewer 1 Comment 11: As this is a protocol paper, there are no results yet. Consider adding a section describing the expected results/analysis.

We believe we already incorporated our expectations into the manuscript, namely in line 70: Therefor it can be anticipated that particularly patients with low SMM might benefit from weekly low dose cisplatin based concurrent CRT to achieve an adequate cumulative dose comparable to patients with normal SMM’ and in line 273-278: ‘By selecting a subgroup of patients that are especially prone to CDLT, namely patients with low SMM, and randomizing these patients between the triweekly and a the weekly regimen of cisplatin, we believe we can investigate whether the weekly regimen might be this solution. The results of this trial can lead to a change in guidelines for cisplatin treatment, might improve our understanding of the effect of SMM on CDLT and enhance personalized treatment in patients with HNSCC’. 

12. Reviewer 1 Comment 11: The discussion provides good context and rationale for the study design choices.

Thank you, we did not change this part.

13. Reviewer 1 Comment 11: References are appropriate. Limitations of the study design are acknowledged.

Thank you, we did not change this part.

14. Reviewer 1 Comment 11: Splitting some lengthy paragraphs into two would further enhance readability.

Line 181:’.. morbidity.12,39 A systematic..’ we have split this paragraph in two between these sentences. 

Line 211: ‘10 years. In line’ we have split this paragraph in two between these sentences.

Line 228: ‘low SMM vary.17,45 In the CISLOW study’ we have split this paragraph in two between these sentences.

Furthermore we rectified one sentence in the abstract, line 8, since cisplatin 200 mg/m2 or more is thought to be the optimal dose, instead of more than 200 mg/m2. Szturz et al. and Strojan et al. both propose 200 mg/m2 or more to be the minimum optimal dosage of cisplatin. Spreafico et al. proposes more than 200 mg/m2, but they researched a specific sub-group of patients with oropharyngeal HPV+ carcinomas. We have added this information in line 193 by writing: ‘with HPV positive tumors’ and removed ‘of more than 200 mg/m2 for oncologic outcomes, demonstrated by Spreafico et al.(42).’ From line 243.

In line 283 we have added the word ‘is’, to make the sentence correct.

---

## [Editor Report · Decision Letter 1]

26 Oct 2023

Low dose cisplatin weekly versus high dose cisplatin every three weeks in primary chemoradiation in head and neck cancer patients with low skeletal muscle mass: the CISLOW-study protocol

PONE-D-23-16285R1

Dear Dr.

We’re pleased to inform you that your manuscript has been judged scientifically suitable for publication and will be formally accepted for publication once it meets all outstanding technical requirements.

Kind regards,

Antonino Maniaci

Academic Editor

PLOS ONE

Additional Editor Comments (optional):

Well done, the paper is improved. Bests
---

## [Editor Report · Acceptance letter]

13 Nov 2023

PONE-D-23-16285R1 

Low dose cisplatin weekly versus high dose cisplatin every three weeks in primary chemoradiotherapy in head and neck cancer patients with low skeletal muscle mass: the CISLOW-study protocol 

Dear Dr. de Bree:

I'm pleased to inform you that your manuscript has been deemed suitable for publication in PLOS ONE. Congratulations! Your manuscript is now with our production department. 

Kind regards, 

on behalf of

Dr. Antonino Maniaci 

Academic Editor

PLOS ONE